# Motivation and Continuance Intention towards Online Instruction among Teachers during the COVID-19 Pandemic: The Mediating Effect of Burnout and Technostress

**DOI:** 10.3390/ijerph17218002

**Published:** 2020-10-30

**Authors:** Ion Ovidiu Panisoara, Iulia Lazar, Georgeta Panisoara, Ruxandra Chirca, Anca Simona Ursu

**Affiliations:** 1Teacher Training Department, Faculty of Psychology and Educational Science, Bucharest University, 90 Panduri Street, Sector 5, 050663 Bucharest, Romania; ovidiu.panisoara@fpse.unibuc.ro (I.O.P.); ruxandra.chirca@fpse.unibuc.ro (R.C.); anca.simona.ursu@drd.unibuc.ro (A.S.U.); 2Psychology Department, Faculty of Psychology and Educational Science, Bucharest University, 90 Panduri Street, Sector 5, 050663 Bucharest, Romania; georgeta.panisoara@fpse.unibuc.ro

**Keywords:** burnout, COVID-19 pandemic, in-service teachers, motivation, technostress

## Abstract

In-service teachers have various emotional and motivational experiences that can influence their continuance intention towards online-only instruction during the COVID-19 pandemic, as a significant stress factor for their workplace. Derived from the Self-Determination Theory (SDT), Job Demands–Resources Model (JD–R), and Technology Acceptance Model (TAM), the present research model includes technological pedagogical knowledge (TPK) self-efficacy (SE), intrinsic (IM) and extrinsic (EM) work motivation, and occupational stress (OS) (i.e., burnout and technostress which have been examined in tandem) as key dimensions to explain the better continuance intention among in-service teachers to use online-only instruction (CI). Data for the research model were collected from 980 in-service teachers during the COVID-19 outbreak between April and May 2020. Overall, the structural model explained 70% of the variance in teachers’ CI. Motivational practices were directly and indirectly linked through OS with CI. The findings showed that IM has the most directly significant effect on teachers’ CI, followed by TPK-SE, and OS as significant, but lower predictors. IM was positively associated with TPK-SE and negatively associated with EM. The results offered valuable insights into how motivation constructs were related to OS and to a better understanding online instruction in an unstable work context, in order to support teachers in coping during working remotely.

## 1. Introduction

Throughout the past few months, as a result of the COVID-19 lockdowns across the world [1], society has had to face significant challenges, which unavoidably affected the educational field as well, with undesirable effects for teachers working remotely. Online learning has been gaining ground and communication technologies have become an indispensable tool for maintaining work continuity not only in school, but also across all fields of work. With regard to the educational field, these rapid and probably unexpected changes might have generated a series of difficulties in ensuring the effectiveness of both teachers and students [2].

Relative to existing studies, researchers have considered attitudinal and motivational factors as key predictors of continuance intentions for any type of instruction that is accompanied by technology. For instance, Davis Bagozzi and Warshaw [3] proposed several predictors of CI: enjoyment as an example of intrinsic motivation and perceived usefulness as an example of extrinsic motivation. In addition, Ferriz-Valero, Østerlie, Martinez and García-Jaén [4] also mentioned that affective commitment as intrinsic motivation and external regulation as extrinsic motivation are determinants of CI. Moon and Kim [5] found that playfulness has a significant effect on the intention to use the World Wide Web. Thus, in the usual synchronous and asynchronous interaction in an online environment [6], the emotions that teachers feel, such as enjoyment [7,8], affection associated with a digital tool [9], computer playfulness [10] or computer anxiety [11] have been recognized as key antecedents of their continuance intention to use digital technologies. Therefore, researchers have integrated motivational factors to understand the external variables that affect the continuance intention to use digital technologies.

Nowadays, given the high levels of acute stress reactions induced by the current health crisis, it is not surprising that more and more teachers complained about the negative affective response to online technology use. Online environments for teaching and learning can be effective, but only when the conditions in which teachers can successfully continue teaching are known. Motivation is the psychological mechanism of activation for the way in which a teacher acts. This is an external or internal determinant of their behaviors, along with other regulating mechanisms. Therefore, there is an undeniable link between motivation and the continuation of online teaching, be it external or intrinsic, cognitive or affective, positive or negative. The result of all motivational dimensions will affect any professional activity (i.e., teachers’ jobs) in a specific way.

In the online education context, the main attention has been granted to continued usage behavior [12]. Consequently, the continuance of long-term usage was predicted and examined through various constructs, in different settings [13]. It is therefore recognized that there are many variables affecting the adoption and long-term use of online learning.

The educational workplace was predictable and easy to control, at least until the beginning of the present healthcare crisis. In the current pandemic situation, teachers are not able to control the educational environment in which they do their work. There may be certain unknown variables affecting the continuance intention of online instruction, especially in a crisis context, when the role of various emotions is much more pronounced, and the degree of uncertainty about the working environment is high.

Furthermore, limited research has integrated negative and positive feelings related to the long-term usage of online instruction in an unstable work context in one model to explore how these constructs relate to each other and sustain teachers’ continuous interest in online teaching and learning. Moreover, Kim, Chan and Chan [11] stated that only a few studies have taken into consideration emotional factors as predictors of continuance intention. 

This study contributes to the existing literature by expanding the knowledge of teachers’ motivation and continuance intention to use online instruction in an unstable work context, highlighting the direct and indirect negative affective response to technology. The identification of affective variables and the testing of a new theoretical research model, on the basis of which the contribution of both negative and positive emotional factors can be investigated simultaneously in an unpredictable work setting, would fill a gap in the literature and provide a model for forecasting teachers’ continuance usage of online instruction. Consequently, the present research contributes to our understanding of the complexity of teachers’ emotions in the workplace by evaluating the relationship between emotional and motivational factors and the continuous use of online instruction in an unforeseen setting simultaneously, which is not evident in the extant literature. Therefore, a research exploration based on the Self-Determination Theory (SDT), Technology Acceptance Model (TAM), and Job Demands–Resources Model (JD–R) was conducted in the present study to explore teachers’ intention to continue using online instruction in an unstable work context, to support them and help them cope while working remotely. Thus, the primary purpose of the present study was to fill the identified gap, investigating the internal relationships between five first-order constructs (technological pedagogical knowledge (TPK) self-efficacy (SE), extrinsic (EM) and intrinsic motivation (IM), occupational stress (i.e., burnout and technostress (OS)) as variables of the structural model to explain teacher intention to continue instruction online (CI). This evidence should inspire further efforts to discover new predictors that may encourage teachers’ continuance intention to use online instruction, even in an uncertain working environment, as well as their impediments to accepting it. 

Accordingly, the present study focuses on the following research questions:

Q1. Are there links between TPK self-efficacy, extrinsic and intrinsic work motivation, occupational stress (i.e., burnout and technostress), and continuance intention to use online instruction among in-service teachers? 

Q2. Are there any associations between motivational dimensions among in-service teachers?

Q3. Does occupational stress (i.e., burnout and technostress) mediate the relationship between motivational factors and continuance intention to use online instruction among in-service teachers?

Understanding the mediating roles of occupational stress with technology use, and examining these specifically in the context of links between factors of the structural model, will contribute to shaping post-adoption continuance intentions to use online instruction by in-service teachers, specifically in the present context of the extension of online instruction due to the global pandemic caused by the coronavirus disease.

## 2. Theoretical Framework and Hypothesis

### 2.1. Development of Conceptual Model

This research explored the effect of key cognitive–affective factors on the continuance usage intention of online instruction in an unstable workplace. The recommended research model of the present study integrated significant elements of motivational issues and job-specific demands caused by the COVID-19 pandemic within the critical context of a school, as well as their interactions, as the main factors underlying teacher continuance intention to accept technology (Figure 1). 

The current studies on teachers’ continuous online instruction intention are mainly based on several theories and models, such as Self-Determination Theory (SDT) and the Technology Acceptance Model and its extensions, the Information System, the Theory of Planned Behavior and the Expectation–Confirmation model [9,10,11]. These theories and models have dominated the investigation on the acceptance of digital technologies by end users and have provided great knowledge in relation to this research topic. Self-Determination Theory (SDT) can be perceived as a multidimensional theoretical paradigm that shows the results of interactions between cognitive and emotional factors in complex learning settings [14]. Most of these research models resulted from the Self-Determination Theory (SDT) proposed by Deci and Ryan [15,16], which differentiates controlled motivation versus autonomous motivation. Therefore, the SDT has been widely used to investigate users’ intrinsic and extrinsic motivation as key constructs in a post-adoption situation [17], such as lecture attendance [18], using blogs to learn [19] or using social networking sites [20]. The Technology Acceptance Model (TAM) developed by Davis [21] proposed the key predictors for technology acceptance. Furthermore, TAM has been extended by the addition of other dimensions such as computer self-efficacy, which is also used in SDT or subjective norm to predict usage intention [22]. An overstated effect of extrinsic motivations on the acceptance of technology was observed, while intrinsic motivations were ignored [23]. Job Demands–Resources Model (JD–R) is different from SDT and TAM, even if job demands are common variables and represent a “path of thinking about how job attributes may affect worker health, wellbeing, and motivation” [24]. JD–R was used in most of the previous studies to investigate the negative outcome variables, such as burnout, stress, and poor health [25]. However, no previous research has used the JD–R to explore the instant and concurrent effects of teachers’ burnout and technostress [26] on their behavioral intention to use online instruction. To date, a significant body of literature has focused on examining SDT, TAM, and JD–R independently in clarifying different effective classroom practices in an online environment, but no research has theoretically linked these models thus far. 

Previous studies have rarely focused on understanding the impact of simultaneous actions of burnout and technostress as explicit and correlated responses. Thus, the present study used the research model as it has been theorized to examine the simultaneous contributions of motivation and occupational stress on continuance intention in an unstable workplace. 

Briefly, based on the literature review, the present research examined relationships through TPK self-efficacy, extrinsic and intrinsic motivation based on the Self-Determination Theory (SDT), occupational stress based on the extended Job Demands–Resources Model of Burnout (JD–R), and teachers’ continuance intention to use online instruction based on the Technology Acceptance Model (TAM). Extrinsic motivation is a common predictor used in all three theories and TPK-self-efficacy is a common predictor used in TAM and SDT, as can be observed in Figure 1.

### 2.2. Self-Determination Theory (SDT)

In the theory of self-determination [27], various forms of motivation based on different reasons or goals were recognized. There is undoubtedly ongoing interest in the study of motivation [28] from a multidimensional perspective, delineating, at the same time, the intrinsic side, which implies an approach to objectives caused by the belief that people will feel good, and from the extrinsic one, caused by conjectural reasons for interest [29,30,31]. The SDT constructs used in this research are described below.

#### 2.2.1. TPK Self-Efficacy

Self-Determination Theory (SDT) involves self-efficacy, a similar concept to perceived competence [32] and one of the most used predictors of motivational models for e-learning [33]. Self-efficacy is an important dimension to understand e-learning continuance intention in the workplace [27]. Teachers’ self-efficacy, regarded as the correct perception of one’s own abilities and an interest in developing students’ competencies, is known to positively influence students’ performance, involvement, wellbeing, attitude towards school, and academic success [34]. Concurrently, teachers’ self-efficacy has been proved to be a predictor of higher engagement and lower burnout [35], which advocates the complementarity of the factors investigated in this study. A study revealed that an 8-month computer training program for teachers, in terms of its integration into education, significantly reduced teacher computer anxiety [36]. This indicates that teachers’ level of knowledge regarding the use of technology influences their anxiety about using it. Another study showed that TPK plays a key role in reducing teachers’ technostress, which suggests that improving teachers’ TPK abilities, through school support and an increase in computer self-efficacy, is essential. Teachers need to employ multiple sources of knowledge, consider various contextual factors and find adequate means to test their lessons’ feasibility [37]. On the other hand, teachers that consider themselves to be more digitally efficient and better institutionally supported experience stronger positive emotions when they use digital resources in their classes, and they are more motivated and autonomously employed in their activity [38]. 

Therefore, self-knowledge and knowledge of teachers’ negative affective responses to technology use, like burnout and technostress levels, followed by self-evaluations of the effectiveness of online teaching, are possible predictors of the intention to continue using online teaching. It may be fruitful for educational research to observe the effect of context on the self-efficacy of technology integration and difficulties in the adoption of interactive teaching and learning [39] as well as the motivational and affective attitudes of in-service teachers. Authors use various terms when exploring the ability and competence of a teacher to use technology for educational purposes, such as technology integration self-efficacy [40], computer self-efficacy [41], or information and communication technologies (ICT) self-efficacy [38]. The present research embraced the term TPK self-efficacy for this dimension and proposed to investigate the complementary effects of motivations and affective responses in relation to online instruction continuance intention by in-service teachers.

From this perspective, the research was performed to test the following direct effects hypotheses:
**Hypothesis** **1.***H1 TPK self-efficacy is positively and directly related to continuance intention to use online instruction*.
**Hypothesis** **2.***H2 TPK self-efficacy is negatively related to burnout and technostress*. 


#### 2.2.2. Intrinsic Motivation

According to SDT, intrinsic motivation can be perceived as a determinant of action in which a person became involved due to their interest [42], preceded by intentions such as continuance intentions to use e-learning [43]. In the educational field, intrinsic motivation is highly respected due to its consequences, which have an immediate effect: “motivation generates” [31]. Intrinsic motivation is usually correlated with positive employee results, and extrinsic motivation expressed by external regulation is negatively correlated with or unrelated to positive results [31,44]. One pragmatic suggestion is that organizations should focus on intrinsic and extrinsic motivations as separate predictors that influence different outcomes [44]. Along with intrinsic and extrinsic motivation, self-efficacy perceptions of technological pedagogical knowledge (TPK) [38,45] can promote the continued usage of e-learning [22]. Good knowledge of intrinsic motivation is an essential step in increasing a teacher’s capacity to adapt to crisis situations, which has the effect of reducing technocracy and, implicitly, burnout. Less perceived intrinsic motivation can thus become a source of anxiety reduction. Therefore, there is constant interest on the part of researchers in understanding the predictors of intrinsic motivation and explaining how they are constructed, modified, and structured. 

#### 2.2.3. Extrinsic Motivation

Motivation is the psychological regulatory mechanism that refers to “the dynamics of behavior, the process of initiation, support and direction of an individual’s activities” [46]. Extrinsic motivation is met when a person performs an action to fulfill social expectations, while simultaneously avoiding sanctions or complying with external control [47]. Extrinsic motivation, involving external regulation, occurs when “behaviors are controlled to obtain a reward or to avoid a constraint” [48], and are positively influenced by individual and contextual antecedents [22]. In the present research, extrinsic motivation is linked to the obligation of authorities to teach exclusively online during the COVID-19 pandemic. 

From this perspective, the research was performed to test the following hypotheses for: (a)direct effects
**Hypothesis** **3.***H3 Intrinsic motivation is positively and directly related to continuance intention to use online instruction*.
**Hypothesis** **4.***H4 Intrinsic motivation is negatively related to burnout and technostress*.
**Hypothesis** **5.***H5 Extrinsic motivation is positively related to continuance intention to use online instruction*.
**Hypothesis** **6.***H6 Extrinsic motivation is positively related to burnout and technostress*.(b)correlation power among exogenous variables
**Hypothesis** **8.***H8 Intrinsic motivation is positively associated with TPK self-efficacy*.
**Hypothesis** **9.***H9 Intrinsic motivation is negatively associated with extrinsic motivation*.



### 2.3. Job Demands–Resources Model of Burnout

The relationship between job-specific demands and outputs within the educational context assumes that “employee health and wellbeing result from a balance between positive (resources) and negative (demands) job characteristics” [24]. An extended JD–R model, which included the performance dimension [49], described burnout as something that “results from high work demands and poor job resources” [24]. High job burnout among in-service teachers caused by the current global health crisis, which has influenced educational activity, may be driven in part by their ability to become familiar in an extremely short timeframe with the diversity of technology-enhanced teaching forms, alongside the demands of working exclusively remotely using different digital tools that can generate technostress. 

In the present research, burnout and technostress have been examined in tandem and were consequently treated as a unitary instead of a two-dimensional construct, as derived from an exploratory analysis. The JD–R constructs used in this research are described below.

#### 2.3.1. Burnout

Burnout is a multidimensional construct encountered by people who reach the final stage of chronic occupational stress. Maslach and Jackson define burnout as a “syndrome of emotional exhaustion, depersonalization and reduction of personal achievement that can occur among people who “work with people” of any kind” [50]. Initially, the term “burnout” was applied to those working with beneficiaries (e.g., students, customers, or patients), but this concept was later used for other professions as well, such as educational instructors [51,52]. Specialty studies have identified multiple factors that can lead to teacher burnout, such as low levels of social interaction, which are self-evident within the current pandemic context [53]. Consequently, the large amount of research conducted worldwide suggests the existence of a global interest in studying teacher burnout [54]. These studies revealed that burnout is a work-related phenomenon [55], which influences the quality of teacher performance, such as work efficiency [56], interpersonal relationships and overall teacher wellbeing [57]. Specifically, if teachers’ expectations do not correspond to the reality of their workplace, this can lead to multiple forms of exhaustion: emotional, physical, spiritual [58]. Based on these studies, there several theoretical models and tools have been developed for assessing teacher burnout, a concept which refers to a negative mental state related to work, which affects psychological health [59], job satisfaction [60] and teachers’ wellbeing [61], as well as their students’ academic performance. Another study on in-service teachers revealed that their state of exhaustion is due to emotional exhaustion rather than to low personal achievement or depersonalization [62]. Teachers who possess more personal resources and skills cope better with job challenges and are less prone to burnout [63], while those with high degrees of neuroticism, perfectionism, and a desire to help others are more vulnerable to burnout [64]. 

With regard to the organizational environment, it has been found that teachers’ level of burnout is associated with their relationship with the principal or his leadership style [32], with the level of trust in their school [65], and with certain types of school cultures and environments [32]. Other research suggests that social interactions within the school [66] can be a source of teacher burnout, and these interactions may vary depending on certain work contexts within the same school. People are different and they have distinct beliefs, personality traits, thinking styles, and socio-emotional skills, that can moderate the negative effects of organizational stressors. Considering that teachers are expected to meet their students online daily and to collaborate with parents, administrators, counsellors and other teachers, certain potentially stressful interactions might be generated. In particular, in the context of this pandemic, teachers are required to constantly adapt to changes in the system, which can sometimes be radical, and to encourage their students to meet high educational standards. Consequently, it is not surprising that, during their professional experience in the online environment, many teachers face burnout. Longitudinal studies of burnout suggest that its prevalence among teachers may be increasing, especially now due to the decrease in teachers’ confidence in their relationships with administrators, colleagues, parents, and students. This decrease in their confidence may be due to the different responsibility standards or other external factors, such as the current global health crisis, which have considerably changed the teaching profession [67]. 

Briefly, significant changes in the education system resulting from the obligation to teach exclusively online can lead to the development of mental health problems within the teaching profession [60,68]. Teachers have to cope with many stressors, such as unusually high workload in relation to preparing lessons, the ambiguity of the role, or difficulties in class management in the virtual environment [69,70,71,72]. Consequently, in the present research, burnout is linked to the exhaustion related to teaching exclusively online during the COVID-19 pandemic.

#### 2.3.2. Technostress

Technostress was defined as a problem of improper adaptation caused by the failure of people to cope with technology and the changes in requirements related to the use of technology, which generate psychological and physical stress towards the latter [73,74]. The role of teachers is essential in integrating technology into education. This is the key to the adoption of technology, and one of the significant problems of its integration in the field of education is its acceptance by teachers [75]. Although the use of technology in education is encouraged, some studies identify many obstacles in achieving it, such as a lack of training, inadequate infrastructure, and a lack of support from technology specialists, etc. These obstacles can induce teachers’ anxiety and tension, leading to mental and physical stress related to the use of technology [76]. Even if the effective use of information and communication technologies (ICT) in education improves teaching and assessment and also student performance, the skills and knowledge requirements of teachers to effectively use ICT can increase at the same time, creating additional workload, challenges and stress. They are continually trying to keep up with the evolution of technology and the innovations associated with pedagogy at the same time. This can lead to increased stress levels in teachers, who are forced to adapt to changes [77]. 

Research on technostress has focused more on industrial and government sectors and less on education. Research in the educational field has instead considered university teachers, as they are the first to use technology more predominantly, and to bring innovation into education while having to continuously update their knowledge and skills and, as a result, they face a large amount of job pressure [78]. The lack of training, the inadequate infrastructure and the lack of technological support generated by mandatory online teaching can increase teachers’ anxiety and tension, which lead to mental and physical stress related to the use of technology [78]. 

As a result of multiple studies, researchers discovered five categories of factors that induce technostress (technological overload, technological invasion, technological complexity, technological insecurity and technological uncertainty) and three inhibitors of technostress, namely literacy facilitation, providing technical support provision and involvement facilitation [79]. A qualitative study was conducted to test whether teachers’ technostress can be induced by the discrepancy between their abilities, their needs and the available technological support (e.g., training and technical assistance) [80]. Technostress has been reported by users of digital resources with different degrees of difficulty—for instance, learning management systems (LMS), digital data processing devices, social media, collaboration tools and mobile applications [78]. Other variables were also related to technostress, such as teaching experience and the level of teachers’ psychological capital. It was thus concluded that technostress is an important predictor of the psychological capital and that a decrease in the technostress level will increase the level of psychological capital [81]. 

Previous studies reported that excessive technostress can have negative consequences for individuals regarding their personal life, their physical and psychological health (i.e., depression, concentration difficulties and social/relationship problems), as well as their professional life (i.e., decreased job satisfaction, reduced organizational commitment and low job performance) [82]. Other possible causes reported by teachers were the extra time needed to prepare online classes, unexpected errors, low technological reliability, and a lack of training in the use of technology. As a result, teachers were confronted with multiple negative symptoms, such as headaches, fatigue, sadness, or nervousness. Technostress was negatively associated with job satisfaction and performance [74,83]. Nonetheless, technostress is not only a teacher health problem, but also an organizational management problem [76]. 

To conclude, technostress currently represents a major problem in the educational field. The social and institutional pressure faced by teachers regarding the integration of technology, as well as the lack of information and support, determine whether teachers will experience technostress. These negative emotions experienced by teachers regarding the use of technology for online teaching can prevent teachers from continuing to use technology and can significantly influence teaching and learning activities and, implicitly, students’ performance. In the present research, technostress is linked to teachers’ failure to cope with the challenges of using digital resources to teach exclusively online during the COVID-19 pandemic.

From this perspective, our study was performed to test the following hypotheses for:
(a)direct effects
**Hypothesis** **7.***H7 Burnout and technostress is positively related to continuance intention to use online instruction*.(b)mediated effects
**Hypothesis** **10.***H10 Burnout and technostress mediated the relationship between SDT self-efficacy and continuous intention to use online instruction*.
**Hypothesis** **11.***H11 Burnout and technostress mediated the relationship between intrinsic motivation and continuous intention to use online instruction*.
**Hypothesis** **12.**H12 Burnout and technostress mediated the relationship between extrinsic motivation and continuous intention to use online instruction.


### 2.4. Technology Acceptance Model (TAM)

The Technology Acceptance Model (TAM) suggests that two attitudinal beliefs, namely perceived ease of use (i.e., intrinsic motivation) and perceived usefulness (i.e., extrinsic motivation) [84], are the best predictors of actual system use and for understanding the behavioral intention over time [85]. Nevertheless, researchers indicated the extension of the TAM model with additional factors to provide a stronger model for particular tasks [86]. Because of its simplicity and parsimony [19], TAM has been extensively used to predict the continuance intention to use different digital technologies such as Massive Open Online Courses (MOOCs) [87], mobile Internet services (MIS) [84] or Facebook [88]. The main TAM construct (continuance intention) used in this research is described below. 

#### Continuance Intention

Continuance intention is a key factor in intuiting teachers’ post-acceptance behavior [89] as an important educational cognitive choice [90]. Accordingly, researchers have explored the intention to continue online learning among teachers in different settings, taking into account their experience [91,92], behaviors [93], skills [94], emotions [95,96], gender [97], age [98,99,100] and attitudes [101]. Nevertheless, studies on the continuance intention to use online education are still limited [13,102]. Only a few investigations have examined self-efficacy [103], perceived competences [27,104], and e-learning contexts [105] as predictors of continuance intention to use online learning or self-efficacy as a predictor of exhaustion [106]. In addition, the literature on the behavioral intention to use virtual technologies focuses mainly on perceived satisfaction [13,107,108], motivation [109], the effectiveness of e-learning, engagement and learning outcomes [110], while ignoring negative emotions (e.g., exhaustion, technostress). 

Thus, understanding the issues that obstruct the continuation of using online learning is essential for teachers and educational organizations to survive in a severely competitive educational environment, especially the context of the health crisis occurring all over the world [89]. The research hypotheses that forecast statements about the possible results are represented in Figure 2.

## 3. Method

### 3.1. Questionnaire

Based on the research context and hypotheses, a questionnaire was developed in the format of a seven-point Likert scale. The initial version of the questionnaire was prepared and revised through the use of educational expert panels and included self-developed or previously validated items to ensure content validity [19]. The authors obtained explicit approval to use validated items from other studies (Appendix B). Two English teachers helped to translate items from English to Romanian and later to translate them back into English in the case of adapted items. Six initial constructs from prior studies were adapted based on the research context as the quantifiable dimensions in the current study: TPK self-efficacy, extrinsic motivation, intrinsic motivation, burnout, technostress, and teachers’ continuance intention to use online instruction. The questionnaire, which evaluated teachers’ perceptions related to motivation, occupational stress and continuance intention, used a seven-point Likert-type scale ranging from strongly disagree (1) to strongly agree (7). The instruments used were adapted from the Work Tasks Motivation Scale for Teachers (WTMST) [48], the Oldenburg Burnout Inventory (OBI) [111], the Person–Technology-Enhanced Learning Misfit (P–TEL) Scale [83], and the Continuance Intention Scale (CI) [83,87,105] to measure intrinsic and extrinsic motivation, burnout, technostress and continuance intention to use online instruction. For TPK self-efficacy, items were modified from the work of [112,113]. Items proposed by Fernet and collaborators [48] were used to characterize and operationalize the intrinsic and extrinsic constructs. Items used to shape burnout were adapted from the work of [111], and technostress items were adapted from the work of [83]. To operationalize the teachers’ continuance intention to use online instruction, items were modified from the original works of [83,87,105]. Before starting the research, permission to partially reproduce survey questionnaires or psychometric scales was obtained. 

The first part of the questionnaire presented general information about the study (i.e., title, purpose, who is invited to participate, the ethics related to the research, etc.). In the second part of the questionnaire, the following specific information about the voluntary participants’ profiles was requested: age, gender, school setting, the degree of education obtained by continuing training, and the length of the term of employment. The third part of the questionnaire contained closed-ended questions related to the body content of the research. The Commission of Ethics and Academic Professional Deontology of the University of Bucharest (UB) approved the use of the questionnaire after reviewing it. The list of measurement items corresponding to the six initial factors and their sources is presented in Appendix B.

### 3.2. Data Collection and Participants

Educational institutions in Romania were obliged by the legislation adopted in the emergency regime to suspend traditional teaching on 16 March 2020. Teacher–student and student–student interactions were interrupted and teaching face-to-face was replaced with teaching online. The research sample focused on in-service teachers from Romania who had an active connection to the Internet, who were teaching online and who were interested in participating voluntarily in the study. Google Forms was the instrument used to request their answers. 

This research was performed using a split-sample model development and a subsequent model cross-validation strategy [114]. According to Young and Pearce [115], “Exploratory Factor Analyses (EFA) should be followed by Confirmatory Factor Analyses (CFA) using a different sample (or samples) to evaluate the EFA-informed a priori theory about the measure’s factor-structure and psychometric properties”. The first stage of sampling was conducted from 4 April 2020 to 14 April 2020, when the state of emergency was extended, and the second stage was performed from 15 April 2020 to 9 May 2020. Data were collected using a web-based questionnaire that was posted in targeted Facebook groups from 4 April to 9 May 2020. Several rounds of questionnaire distributions were necessary to increase the response rate. The valid sample included 980 in-service teachers, after excluding seven invalid responses (see Appendix A). Among the voluntary teacher participants working at schools in Romania, 11.2% were early stage, 16% were definitive, 15.9% had earned a second-degree certificate, 51.5% had a first-degree certificate, and 5.3% had earned a Ph.D. degree. A total of 857 (19.6%) taught in an urban environment, and 123 (12.6%) taught in a rural environment. There were 949 female participants involved (96.8%) and 31 (3.2%) male participants, aged between 20 and 68 years. Eight hundred and ten (82.7%) of the participants were employed for an unspecified duration, and 170 (17.3%) were employed for a fixed duration.

### 3.3. Statistical Analysis

Firstly, the analysis of the principal components using Exploratory Factor Analyses (EFA), which includes the main information of the measured data in such a way as to be able to highlight the similarities and differences between them, was performed easily. In brief, the purpose of the principal components is to extract a small number of linear combinations of the independent components from a set of measured variables that retain as much information as possible from the original variables [116,117]. Standard methods bias, the primary source of measurement error [118], was also considered using Harman’s one-factor approach, which recommended the total variance for one factor up to 50% for the absence of a common method bias [119]. For the descriptive statistics and EFA, the SPSS software was used. 

The reliability (i.e., Cronbach’s alpha (α), composite reliability (CR)) and validity of each construct (i.e., the average variation extraction (AVE), the maximum shared variance (MSV)) and the maximum reliability (MaxR (H))) was evaluated using SPSS and AMOS software. The test for the reliability, convergent validity and discriminative validity involves an assessment of the degree of consistency between several measurements of a variable that is required before the primary processing of experimental data [117]. Convergent validity means that all items have the ability to measure the same construct. Cronbach’s alpha coefficient (α) was used to test the internal consistency of the study performed. If the values of this coefficient exceed the value of 0.90, then reliability is excellent; reliability is high between 0.70 and 0.90; reliability is moderate between 0.5 and 0.7 [120]. For each construct, the composite reliability parameters (CR) and (AVE) were calculated to measure the validity of the structure. The indices testing the convergent validity are the coefficient of convergent validity (CR) and the coefficient of the Average Variance Extracted (AVE), as stated by Lee [121]. If both conditions are met at the same time (CR > 0.70 and AVE > 0.50), it can be specified that the study has an appropriate converging validity. Different validities mean that, although all items converge to the same size, they nevertheless measure different inputs of the factor [122,123]. Briefly, we examined to what extent the measurement model fit the set of observations.

Structural equation modelling (SEM) was used for its ability to assess the direct, indirect, and mediating relationships between constructs, as estimated in the proposed model [124]. The regression coefficient, i.e., beta (β), a test of path significance (*p*), and the coefficient of determination (R2) values indicate the predictive ability of the model. R^2^ values of 0.20 and higher indicate substantive influence [19]. The model fit indices used to measure the absolute fitness were the following: chi-square normalized by degrees of freedom (chi-square/df) with a maximum cut-off of 5.0, root mean square error of approximation (RMSEA) with a suggested value estimate of 0.08, the comparative fit index (CFI) with a suggested value estimate of 0.90, the Tucker–Lewis index (TLI), with a suggested value estimate of 0.90 and the standardized root mean square residual (SRMR) with a suggested value estimate of 0.08 [125,126].

## 4. Results

### 4.1. Dimensionality Results

The first version of the questionnaire comprised six factors and 36 items. Overall, the questionnaire’s internal consistency was high (Cronbach alpha α = 0.858). Therefore, the alpha values were described as excellent (α = 0.929) for technostress, for TPK self-efficacy (α = 0.928) and intrinsic motivation (α = 0.917), high (α = 0.830) for burnout and continuance intention (α = 0.876), and moderate (α = 0.680) for extrinsic motivation. This research was performed using a split-sample model development and a subsequent model cross-validation strategy [114]. The first stage of sampling (*n*_1_ = 462 (47.1%) voluntary participants) was conducted to explore the preliminary assessment of scale one-dimensionality based on Exploratory Factor Analyses (EFA) [127]. 

The results from the EFA did not confirm the proposed six-factor scale to test the continuance intention to use online instruction. Instead, the EFA output indicated five subscales: TPK self-efficacy, extrinsic motivation, intrinsic motivation, burnout, technostress, and teachers’ continuance intention to use online instruction. The five-factor solution was due to the burnout items mixing with technostress items. 

Consequently, the corresponding factor to burnout and technostress items was labeled burnout and technostress. The five-factor solution cumulated 65.352% of the total variance. There was no common bias, according to Harman’s single-factor test [128]. After the Promax rotated factor, the first dimension of the scale, labeled TPK self-efficacy, explains 37.246% of the total variance (and the second dimension of the scale, labeled burnout and technostress, explains 16.589% of the total variance). The value of the Cronbach’s alpha coefficient corresponding to the entire scale found was 0.923, demonstrating that the applied survey has very good internal consistency [129]. The results of the EFA analyses showed that the value of KMO was 0.949, and Bartlett’s sphericity test was significant [130]. Factor loadings as results of EFA are presented in Table 1. After EFA, one non-adaptive item was eliminated.

### 4.2. Measurement Model Results

The maximum likelihood estimations method for each construct was used to perform Confirmatory Factor Analysis (CFA) [131]. The latent factors of the EFA model were validated by confirmatory factor analysis (CFA) [132] using the sample corresponding to the second stage (*n*_2_ = 518 (52.9%) for voluntary participants). The fit indices for the measurement model are chi-square χ^2^(df) = 2.851, comparative fit index (CFI) = 0.944, Tucker–Lewis index (TLI) = 0.933, root mean square error of approximation (RMSEA) = 0.060 and standardized root mean square residual (SRMR) = 0.064 [133]. Additionally, the CFA model was used as a tool to test the construct validity of the scale. The results of CFA model with five subscales are presented in Figure 3.

The structural validity (convergent and discriminant validity) was confirmed by the results obtained (Table 2) [121]. The convergent validity is confirmed because the CR test result is greater than 0.70 and the AVE test result is greater than 0.50. The discriminant validity is confirmed because each construct’s AVE is greater than the squared correlation of the construct with any other construct [127] (Table 2). After analyzing the results of the convergent and discriminant validity test, 30 items were finally selected for the five-factor solution. Five items were eliminated during CFA due to high values in the standardized residual covariances. Table 2 illustrates the validity of the constructs.

### 4.3. Structural Model Results

To test the adequacy of the proposed structural model, three criteria were used [134]—model fit indices, (2) standardized path estimate significance (*p*), and the amount of variance (R^2^)—which show the predictive power of the model [19] explained in each of the endogenous dimensions (i.e., burnout and technostress and continuance intention). SEM is an ideal method since it allows us to test the mediating effects between endogenous and exogenous dimensions. 

The overall goodness of fit of the research model was acceptable according to all fit indices: chi-square normalized by degrees of freedom (chi-square/df) (χ^2^/4 = 2.059, *p* = 0.128); RMSEA = 0.045; CFI = 0.999; TLI = 0.994; SRMR = 0.046. Consequently, the hypothesized relationships within the structural model were suitable for examination. The findings resulting from the structural model (Figure 4) are presented in Table 3.

### 4.4. Path Analysis Results

Path analysis, using standardized path coefficients, was performed to test the hypothesized relationships. The results of the path analyses are presented in Table 3. As can be seen in Table 3, most of the path coefficients of the structural model are significant. Most of the hypotheses are confirmed. However, hypotheses H2, H5, and H10 are not supported. The structural model accounted for 70% of the variance in teachers’ continuance intention to use online instruction and for 51% of the variance in teachers’ burnout and technostress. 

Teachers’ continuance intention to use online instruction was found to be positively and significantly directly influenced by intrinsic motivation (β = 0.488; *p* < 0.001), which confirms hypothesis H3, and TPK self-efficacy (β = 0.435; *p* < 0.001), which confirms hypothesis H1, burnout and technostress (β = 0.059; *p* < 0.05), which confirms hypothesis H8, but not directly by extrinsic motivation (β = 0.001; *p* > 0.05), which does not confirm hypothesis H5. Moreover, burnout and technostress were significantly influenced by intrinsic motivation (β = −0.364; *p* < 0.001), which confirms hypothesis H4 and extrinsic motivation (β = 0.482; *p* < 0.05), which confirms hypothesis H7, but not by TPK self-efficacy (β = −0.051; *p* > 0.05), which does not confirm hypothesis H2. The direct positive association between intrinsic motivation and TPK self-efficacy was significant (β = 0.756; *p* < 0.001), which confirms hypothesis H8, while the direct negative association between intrinsic and extrinsic motivation was significant (β = −0.332; *p* < 0.001), which confirms hypothesis H9.

Finally, the burnout and technostress construct was found to have a mediating role between intrinsic motivation and continuance intention (β = −0.023; *p* < 0.05), which confirms hypothesis H11, and between extrinsic motivation and continuance intention (β = 0.041; *p* < 0.05), which confirms hypothesis H12, while there were no influences between TPK self-efficacy and continuance intention (β = −0.003; *p* > 0.05), which does not confirm hypothesis H10. 

Briefly, nine hypotheses were supported (H1, H3, H4, H6, H7, H8, H9, H11 and H12), while three were not supported (H2, H5, and H10) in our mixed model based on the Self-Determination Theory (SDT), Job Demands–Resources Model (JD–R) and Technology Acceptance Model (TAM).

## 5. Discussions

A thorough investigation of the relationship between teachers’ intrinsic and extrinsic motivation, perceived TPK self-efficacy, as well as teachers’ negative affective responses to technology use and continuance intention to use online instruction in an unstable work context becomes imperative for identifying appropriate methods to explain these issues. This research aims to estimate the role of burnout and technostress as the main dimensions of occupational stress, and their interactions with other endogenous and exogenous latent variables of the research model in the context of the COVID-19 pandemic, between April and May 2020, using data from a sample of 980 Romanian teachers. The model used for the research was composed of five first-order factors (i.e., TPK self-efficacy, extrinsic motivation, intrinsic motivation, occupational stress (i.e., burnout, technostress), and teachers’ continuance intention to use online instruction). This research expanded upon existing theories (i.e., Self-Determination Theory (SDT), Job Demands–Resources Model (JD–R), and Technology Acceptance Model (TAM)) by examining the effects of one major stress source that determined in-service teachers to work online. When teachers perceived themselves as having TPK self-efficacy, they were motivated to work in the imposed conditions. In addition, the perception of self-efficacy directly supported them in continuing the activity, leading to pleasure and good results.

### 5.1. Theoretical Implications

The biggest challenge of instruction during the COVID-19 pandemic [135], which “generated the largest disturbance of education systems in history” [136], is to stimulate the interest of teachers to continue teaching and learning in virtual environments [137]. However, little is known about how different facets of teachers’ motivations predict their behavioral changes regarding their continuance intention to use e-learning under the influences of negative affective responses such as burnout and technostress. 

Most of the past studies distinguished between consequences of organizational stressors, such as work overload, job insecurity or role ambiguity [138] and daily job demands or daily exhaustion [139]. The former, organizational stressors, cause technostress and the latter, daily job demands or exhaustion, cause burnout as an independent and subsequent outcome of technostress [140]. Specifically, the present research results suggest that in the new global health crisis context, Person–Technology-Enhanced Learning Misfit is highly related to burnout and technostress, and these jointly shape the occupational stress factor. 

This study explored the structural relationships between three motivational factors (i.e., TPK-self-efficacy, intrinsic and extrinsic motivation) influencing in-service teachers’ negative affective response to online instruction (i.e., occupational stress consists of technostress and burnout) and their continuance intention to use it. One of the advantages of the present research is the simplicity of the cognitive–affective scale because it covers five dimensions with 29 items. The convergent and discriminant tests were confirmed by the values obtained for the coefficients shown in Table 2. All these results confirm the good psychometric properties of the measurement scale. Moreover, with a multidimensional measure of situational motivation, the present research extends the possibility to report on important theoretical issues. In this framework, the research results (RQ11-3; RQ2; RQ3) show the causal effects hypothesis of the structural model (Table 3), and can be optimally clustered into three main parts (Parts A, B and C) corresponding to the research questions (Q1, Q2 and Q3).

Considering the fact that emotions can be defined as “multidimensional constructs comprising affective, psychological, cognitive, expressive, and motivational components” [141], the present study enriches the relevant literature on emotions in the virtual workplace among in-service teachers from the perspective of the interrelationships and intrarelationships of motivation, occupational stress and continuance intention to use online instruction in the context of the COVID-19 pandemic, as follows.

#### 5.1.1. Part A, Corresponding to Q1: Are There Links between TPK Self-Efficacy, Extrinsic and Intrinsic Work Motivation, Occupational Stress (i.e., Burnout and Technostress), and Continuance Intention to Use Online Instruction among In-Service Teachers? 

Q1 Response (RQ11): the results indicated direct relationships between motivation and occupational stress.

Teachers who do not perceive themselves as digitally efficient and well supported from an institutional point of view experience more intense negative emotions when teaching online and are less motivated and autonomously involved in their work [38]. However, the present results revealed that there is a lack of direct influence of TPK self-efficacy on occupational stress developed in the context of online instruction usage, which is a different result from other studies [32,142,143,144]. This result may be explained by the fact that the study was conducted in a short period of approximately two months in order to highlight the reaction of teachers in an extremely stressful period. Another cause could be the impossibility of measuring only the TPK-SE effect on occupational stress due to the overlap between IM and TPK-SE. 

The current findings revealed that intrinsic motivation significantly reduced occupational stress, but extrinsic motivation represented by external regulation amplified it to a greater extent. Extrinsic motivation occurred due to external reasons [145] and performing online-only instruction constituted a psychological constraint. This result is, to some extent, similar to those obtained by other researchers who have shown that the school context had a significant direct effect on emotional exhaustion, a burnout dimension [142], or on technostress [146]. Moreover, regarding the link between motivational factors and burnout, Fernet and collaborators [32] stated that “changes in burnout are predicted by changes in teachers’ perceptions of school environment and motivational factors” [32]. Overall, the results of the present research are partially new, due to the exceptional health crisis context that has largely influenced the working environment of in-service teachers.

Briefly, the research findings revealed that extrinsic motivation significantly amplifies occupational stress, represented by negative feelings about the use of online teaching (i.e., burnout and technostress), while intrinsic motivation significantly diminishes them, even if it is with a lower intensity, and the perception of TPK self-efficacy does not affect occupational stress.

RQ12: the results indicated direct relationships between motivation and continuance intention.

The link between motivation and continuance intention was investigated because the understanding of factors shaping decision making is important for improving decision making practices among workers [147]. In explaining the role of affectivity, the authors in [148] concluded that “affect plays a more central role in the decision-making process”. In the present study, the intrinsic motivation was a significant predictor of continuance motivation, suggesting that in-service teachers adopt online instruction because they find this method interesting, a challenge for their personal development, pleasant, innovative, creative, and successful. Teachers need to use multiple sources of knowledge (as the TPK framework suggests), consider multiple contextual factors and find ways to test the feasibility of the online lesson they have designed [37]. Perceived TPK self-efficacy strongly motivated the intention to continue to use online instruction, and this is in line with other results [19,23]. Moreover, intrinsic motivation deeply influenced the intention to continue to use online instruction, and the results were similar to others [149]. By contrast, previous findings have described that intrinsic motivation “does not have significant influence or marginally significant influence on the intention to use computers in the workplace” [134]. Usually, extrinsic motivation (i.e., satisfaction and perceived usefulness) is linked with continuance intention [150]. Contrary to what was expected, external regulation as a form of extrinsic motivation [138] does not directly influence the intention to continue using online instruction. The case analyzed in this research has a constraint, given the context of the research, in the form of an extrinsic motivational determinant, a situation that has not been well investigated thus far. Perhaps this can offer an explanation for our findings. 

RQ13: the results indicated direct relationships between occupational stress (i.e., burnout and technostress) and continuance intention.

During a short period of time, only the strongest emotions are aroused, i.e., the instantaneous effects of stress, and not those developed over time. The high degree of correlation between burnout and technostress is probably justified by the fact that the investigation took place in a short period of time, in which only the exhaustion dimension of the burnout was mainly manifested, with the other burnout dimensions [151] probably appearing later, over time [143]. Additionally, occupational stress may cause people to become less motivated and committed [152] because health, wellbeing and productivity may be disturbed [153]. Even if online-only teaching is a process that clearly involves emotional exhaustion, the present study found only a weak direct influence of burnout and technostress on the intention to continue to use online instruction. Previous researchers also found that the affective components of attitude significantly influence users’ continuance intention in a virtual context [154]. Briefly, the empirical results showed the positive and negative roles of three-dimensional motivational factors on teachers’ intentions to continue to use online instruction, mediated by burnout and technostress. If it is taken into consideration that the presence of a positive and negative overlapping effect, induced by all motivation dimensions related to continuance intention, this result can probably be justified. 

#### 5.1.2. Part B, Corresponding to Q2: Are There Any Associations between Motivational Dimensions among In-Service Teachers?

Q2 Response (RQ2): the results relate to the correlation power between the motivational dimensions. 

Teachers with better TPK self-efficacy for teaching were more intrinsically motivated in their daily tasks. In various contexts, TPK self-efficacy appeared to be strongly linked with intrinsic motivation [155]. Other previous studies have reported similar results [38,156]. On the other hand, the association between intrinsic and extrinsic motivation was observed by many other researchers [157], but the explanations differ significantly from the context in which the online instruction is carried out. Frequently, extrinsic motivation is a motivator, but it is of lower quality than intrinsic motivation [158]. Intrinsic motivation has a stronger effect on continuance intention than extrinsic motivation, which is similar to other studies [159]. 

#### 5.1.3. Part C, Corresponding to Q3: Does Occupational Stress (i.e., Burnout and Technostress) Mediate the Relationship between Motivational Factors and Continuance Intention to Use Online Learning among In-Service Teachers?

Q3 Response (RQ3): the results relate to the power of occupational stress as a mediating factor between motivation and continuance intention.

Finally, the results showed that occupational stress mediated both the relationship between intrinsic and extrinsic motivation and continuance intention, where online instruction usage is mandatory, but the indirect effects have a reduced intensity. The mediating effects may be the result of negative affective responses to e-learning involving trust in the self-capabilities of in-service teachers, which, in turn, increase an individual’s frustration and anxiety [160], which are generated by a limited educational environment imposed by authorities. 

### 5.2. Practical Implications

The increasing popularity of online learning has led to the extensive use of digital resources [161] as additional tools for teaching and learning [93]. The intrinsic motivation represented by the desire to succeed, the pleasure of using this technology and the challenge of finding new and interesting things is negatively correlated with the extrinsic motivation represented by the obligation to teach only online due to the COVID-19 pandemic context. Thus, the influence of intrinsic motivation on continuance intention was slightly diminished by the influence of extrinsic motivation, represented by external regulations. Therefore, many teachers are afraid to make mistakes in the use of online resources [78] or they do not have enough skills and competencies [162] that allow them to rethink the design of virtual learning spaces [163]. In particular, for teachers with less skills related to the knowledge of new information and communication technologies (ICT) [38], problems are intensified by the pandemic. Consequently, the embracing of online instruction could generate additional workload, conflicts, and negative affective responses to technology use for teachers [164]. 

To decrease the negative consequences faced by teachers and generated by occupational stress, it is necessary to accurately identify the predictive factors that can influence the degree of adoption and continuity in the use of technology. The proposed model explains 70% of the continuance intention, which suggests the presence of another 30% of unidentified predictive factors. Based on the research model, predictions can be made to identify, as well as to maintain, the use of online instruction in the future. To succeed, it is necessary to pay more attention to burnout and technostress, which play the roles of latent variables, and which are invisible but of the utmost importance.

The practice of the teaching profession involves continuous professional training. At the time of this study, extrinsic motivational factors were stronger than normal, and continue to be heightened. It has already been recognized that each teacher reacts differently to external sources of stress. Therefore, studying the profile of teachers’ careers is vital for ensuring well-organized educational provision in an online environment, even if the work context is unstable. One example to improve teaching quality is an awareness of the sources that have the greatest impact on the level of technostress (e.g., a lack of time or a large number of students in the virtual classroom who do not have adequate resources for online education) and finding creative solutions to reduce them. Thus, an early awareness of the stress level will lead to better emotional self-control and to the adoption of more effective coping strategies to prevent the presence of burnout dimensions. In addition, the correct management of emotions through the awareness of both positive and negative emotions can help teachers achieve the equilibrium state necessary for effective teaching in unstable contexts. Moreover, an awareness of the motivational strategies adopted by each in-service teacher can lead to better self-control of self-motivation. Consequently, the identification of new descriptors associated with the professional competence standards of teachers in online education, appropriate to their profile (e.g., the development and dissemination of their own teaching materials for differentiated learning in the online environment), can support teachers in their continuous professional development. For education policymakers, the identification of key determinants regarding emotions in the workplace will lead to a better understanding of teachers’ needs in relation to continuous job enhancement. 

Moreover, our research results may be helpful in educational politics, in the sense of promoting reasons for the constraints in the administrative environment, coupled with effective measures to support in-service teachers working remotely.

## 6. Limitations and Suggestions for Future Research

The results have consequences for teachers’ adaptation to unforeseen educational contexts in terms of continued training and education. It is vital to identify teachers’ perceptions about affective and cognitive responses to the use of digital resources and equipment, and about what can be done, from an educational policy perspective, for teachers’ wellbeing. Therefore, this research provides insights into how motivation constructs are related to burnout and technostress to enable a better understanding of online instruction within the COVID-19 context. 

However, there are several constraints in this study. Firstly, having measured simultaneous burnout and technostress, teachers can not realize the difference between emotional exhaustion and stress induced by contextual limitations and the multitude of technical tasks that generate stress related to the technology used. The experimental part of the study was carried out in a relatively short timeframe (i.e., two months), precisely to capture teachers’ emotions and attitudes in a global crisis context. If the study had been extended over a longer period, teachers would certainly have realized the difference between burnout and technostress and could have fought against the opposing psychological phenomenon. 

Another limitation was correlated with the conception of the motivational factor, with more descriptors for intrinsic and less for extrinsic motivation. A possible explanation for this limitation was given by the fact that, during pre-testing, some items of the Oldenburg Burnout Inventory (e.g., “Usually, I can manage the amount of my work well”) were confused with items belonging to the TPK self-efficacy factor. In the preliminary stage of research, items were selected based on the experts’ opinions, who indicated which of these belonged without doubt to the burnout factor. As a result, the number of items dedicated to the burnout measurement was less than the dedicated TPK self-efficacy or stress measurements. Therefore, the research results were limited by the possibility of item selection bias.

The present study was performed using a short-term picture of teachers’ behavior. Consequently, longitudinal examinations are necessary to establish the validity of the hypothesized model and to decide if and how the links among dimensions vary over time. In future research, continuance intention to use online learning in terms of asynchronous and synchronous environments should be considered. Thus, exposure to different teaching models online, via social media, could contribute to a better understanding of the relationship between IM and CI. One solution in the future will be to test other research models with control variables, such as digital age or gender, to discover other interactions and interplays within technology usage.

## 7. Conclusions

The results of this study highlighted significant links between the five cognitive–affective factors in an unstable work context. Intrinsic motivation influences, with a strong positive intensity, the intention to continue online teaching and, with a strong negative intensity, burnout and technostress. Teachers use digital resources and are concerned with knowledge of their field to successfully fulfil their work tasks. They perceived the obligation to teach, and not the intrinsic desire to teach, so a large number of them did not feel psychologically that they were doing this activity because they were able to do it, but because they were obliged to, thus explaining the negative correlation between the two motivational factors. In an uncontrolled workplace context, the only thing a teacher can do is to control how they respond to the stress of imposing online teaching. The first step is to know the possible causes that can generate negative affective responses to technology use. Intrinsic motivation is a more effective process of stimulation and action support than extrinsic motivation because it involves more powerful psychological resources that can sustain and direct the activity until its completion. Nevertheless, in the absence of intrinsic motivation, the continuation of teaching could be achieved only on the basis of extrinsic motivation (i.e., fear of losing one’s job, the need for a salary that ensures basic needs, or stigmatization that they have not fulfilled their tasks). In other words, a teacher’s intrinsic motivation, in association with knowledge related to technology integration, may reduce an individual’s perception of difficulty in relation to online instruction. 

As consequences of the present research, new relationships have been demonstrated to foresee in-service teachers’ continuance intention to use online instruction. The results indicated that the occupational stress variable was strongly affected in a mixed manner: negatively by intrinsic motivation and positively by extrinsic motivation. Moreover, a small indirect influence of burnout and technostress on the link between intrinsic and extrinsic motivation and continuance intention was observed.

Teachers who perceive themselves to be effective and who attribute their success to their personal efforts will find the psychological resources to continue until the completion of an activity. At the opposite end of the spectrum, if they do not perceive themselves as having self-efficacy, online instruction may be stopped by them. Therefore, as in many other adaptive emotional processes, teachers need to become aware of this phenomenon. Any psychological intervention involves awareness and concerted action, in order to replace the old behavior with a new one. Consequently, teachers will find adaptation and adjustment mechanisms more easily and will cope more quickly with psychological phenomena, such as burnout and technostress. 

The present findings may be the blended results of two contradictory influences: one positive and one negative. A deeper investigation of affective causes, coupled with those that can generate positive and negative attitudes, could contribute to increasing the intention to use digital resources in remote instruction. Burnout and technostress are influenced by external conditions, but also by personal coping strategies. Consequently, if coping strategies are formed, problems can be overcome in any crisis. To achieve this goal, extensive mixed research is needed to identify all the elements that can significantly contribute to the support for the intention to continue online teaching in an unfriendly and unusual environment. The results of this study complement the knowledge in the field by introducing the affective dimension alongside the cognitive dimension in a situation that, until now, was completely unusual to all of mankind.

## Figures and Tables

**Figure 1 ijerph-17-08002-f001:**
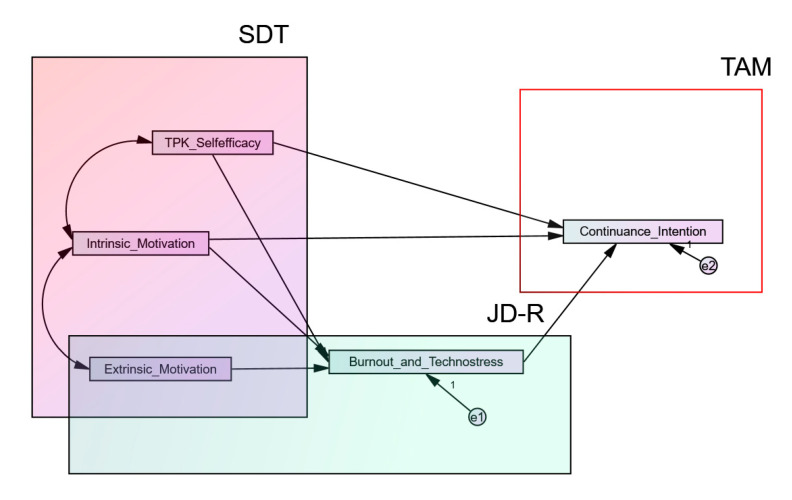
The recommended research model.

**Figure 2 ijerph-17-08002-f002:**
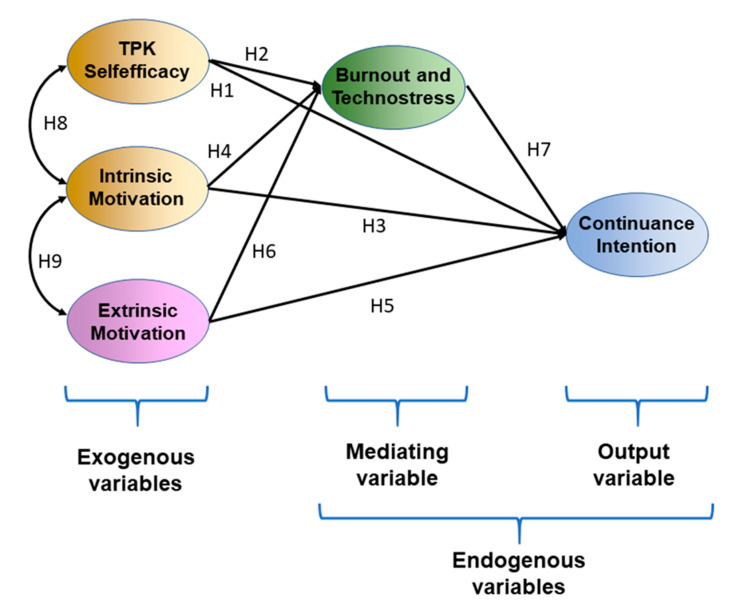
The research hypotheses.

**Figure 3 ijerph-17-08002-f003:**
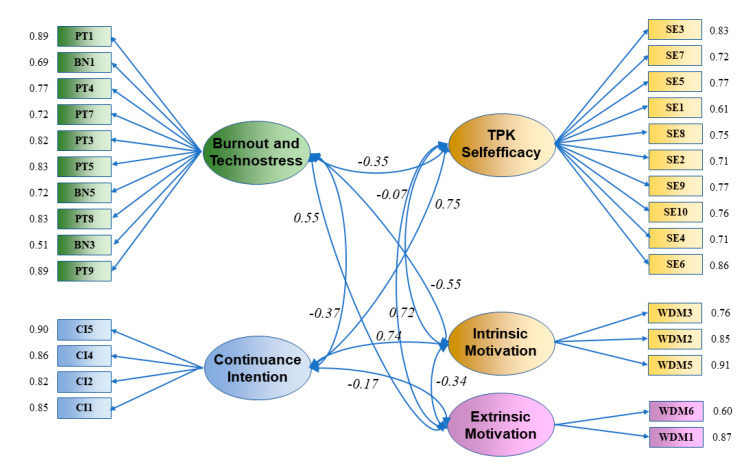
Results of CFA with five subscales. Coefficients mentioned in line with each item are standardized factor loadings. Correlation coefficients between model dimensions are in italics. All coefficients are statistically significant.

**Figure 4 ijerph-17-08002-f004:**
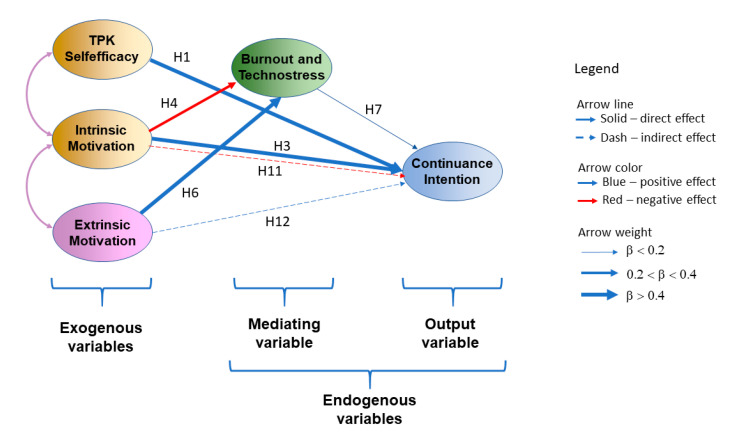
Model of continuance intention to use online instruction. Only significant hypotheses are represented.

**Table 1 ijerph-17-08002-t001:** Factor loadings as results of Exploratory Factor Analyses (EFA) (*n*_1_ = 462).

	TPK ^a^ Self-Efficacy	Burnout and Technostress	Intrinsic Motivation	Continuance Intention	Extrinsic Motivation
SE3	0.908				
SE7	0.888				
SE5	0.817				
SE1	0.792				
SE8	0.777				
SE2	0.714				
SE9	0.696				
SE10	0.691				
SE4	0.657				
SE6	0.632				
PT1		0.820			
BN1		0.819			
PT4		0.790			
PT7		0.776			
PT3		0.764			
PT5		0.745			
BN5		0.743			
PT8		0.728			
PT2		0.720 **			
BN3		0.705			
PT6		0.701 **			
BN4		0.681			
PT9		0.654			
WDM3			0.893		
WDM8			0.822 **		
WDM4			0.781 **		
WDM2			0.726		
WDM9			0.693 **		
WDM5			0.684		
WDM7			0.497 *		
CI5				0.638	
CI4				0.616	
CI2				0.546	
CI1				0.546	
WDM6					0.849
WDM1					0.729

* Eliminated after the EFA; ** Eliminated after the Confirmatory Factor Analyses (CFA); ^a^ TPK signify Technological Pedagogical knowledge.

**Table 2 ijerph-17-08002-t002:** Convergent and discriminant validity coefficients as results of the analysis of structural validity of measurement scale (*n*_2_ = 518).

	CR ^a^	AVE ^b^	MSV ^c^	MaxR(H) ^d^	Burnout and Technostress	TPK Self-Efficacy	Intrinsic Motivation	Continuance Intention	Extrinsic Motivation
Burnout and Technostress	0.927	0.566	0.303	0.940	0.752				
TPK Self-efficacy	0.929	0.568	0.557	0.936	−0.352 ***	0.754			
Intrinsic Motivation	0.880	0.710	0.548	0.899	−0.551 ***	0.719 ***	0.843		
Continuance Intention	0.918	0.736	0.557	0.921	−0.370 ***	0.746 ***	0.740 ***	0.858	
Extrinsic Motivation	0.707	0.555	0.301	0.784	0.549 ***	−0.072	−0.342 ***	−0.170 **	0.745

Note: Significance coding: ^a^ Convergent Validity; ^b^ Average Variance Extracted; ^c^ Maximum Shared Variance; ^d^ Maximum Reliability ** *p* < 0.01, *** *p* < 0.001.

**Table 3 ijerph-17-08002-t003:** The summary of the causal effects’ hypothesis of the structural model.

Results	Predictors	Direct Effect	Indirect Effect	Total Effect	Hypothesis
Continuance intention (CI) (R2 = 0.703 ***)	TPK self-efficacy	0.435 ***			H1 ^c^
			−0.003 ^ns^		H10 ^nc^
				0.441 ***	
	Intrinsic motivation	0.488 ***			H3 ^c^
			−0.023 *		H11^c^
				0.459 ***	
	Extrinsic motivation	0.000 ^ns^			H5 ^nc^
			0.041 *		H12 ^c^
				0.041 *	
	Burnout and technostress	0.059 *	-	0.059 *	H7 ^c^
Burnout and technostress (BT) (R2 = 0.513 ***)	TPK self-efficacy	−0.051 ^ns^	-	−0.051 ^ns^	H2 ^nc^
	Intrinsic motivation	−0.364 ***	-	−0.364 ***	H4 ^c^
	Extrinsic motivation	0.482 ***	-	0.482 ***	H6 ^c^

Note: Significance coding: ^ns^ = not significant, * *p* < 0.05, *** *p* < 0.001; ^nc^ = not confirmed; ^c^ = confirmed.

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
