# Peer review of "Motivation and Continuance Intention towards Online Instruction among Teachers during the COVID-19 Pandemic: The Mediating Effect of Burnout and Technostress"

_ijerph, 2020, doi:10.3390/ijerph17218002_

Round 1
Reviewer 1 Report
Thank you for giving me a chance to review this manuscript. The manuscript has the advantage of understanding motivational factors (internal or external factors) and the influence of self-efficacy on continuous intention through burnout and technical pressure. However, I have noticed some shortcomings that need to be clarified and revised, as listed below:
- The gap and the contribution of the study are not clear. Notably, it is not apparent whether the contribution comes from the theory, psychological factors, and/or the moderator? The theoretical linkage between the motivations and continuous intention is uncertain. Hence, the role of the burnout and technostress has already been confirmed by the prior studies. Specifically, on pages 4, line 65-67: the authors stated that no previous 66 research has used the JD-R to explore the instant and concurrent effects of teacher burnout and technostress on their behavioral intention to use online instruction. What's left here does not postulate adequate contribution to the literature.
- Page 1, line 42, authors stated that “However, while teachers have instantly adopted online instruction, the intention to continue using this type of education is still uncertain”. What do you mean by this type of education? Please rewrite your sentence and clarify the type of education you are discussing.
- Page 1, line 43, authors stated about ‘psychological factors’, I don't know what psychological factor you are referring to in this sentence. Therefore, the author should write sentences carefully to make the reader clearer.
- In addition, research questions (mentioned in introduction) should also be discussed in the discussion section to align with theoretical and practical contributions of this study.
- It should be acknowledged that the methodology and data collection seem rigorous. However, the study severely suffers from theoretical formulation and background.
Author Response
Dear Madam or Sir,
Sincerest thanks to Guest Editor and reviewer for their valuable time and useful contribution.
The authors are grateful to all for careful reading of the paper and helpful suggestions and comments as a way to improve our work. The authors hope that a revised version of the manuscript will still be considered by International Journal of Environmental Research and Public Health.
The authors have modified the manuscript.
Below we provide our responses point by point and modify the manuscript in response to the accurate and intuitive the reviewer’ comments.
We have more clearly highlighted the novelty of the study, explicitly the proposal for a new theoretical research model on the basis of which the contribution of both negative and positive emotional factors can be investigated simultaneously, at the level of the intention to continue using online, in an uncontrolled and unpredictable environment.
We added more studies to the introduction section to clarify the originality and the contribution of research. The study contribution comes from the theoretical research model which included the moderator’s factors.
Therefore, we have supplemented the literature research to better reflect the theoretical aspects and context of the research problem. The introductory part has been fully modified (See Abstract and lines 42-100, corrected version of manuscript). As well we modified discussion and developed limitation and future research (See lines 566-581, 611-613, 677-678, 689-690, 757-766, corrected version of manuscript). Accordingly, with suggestions we modified conclusions (See lines 775-791).
Also, English language was revised.
We hope you find this revision appropriate for research topic. Once more, thank you for your exhaustive review.

Reviewer 2 Report
Dear authors,
Your research is focused on the study of the impact that COVID 19 generated on the sudden use of online teaching.
The sample analyzed is sufficiently representative, 980 active teachers. The statistical treatment is correct.
I would like to know why the issues dedicated to Burnout are less than those dedicated to TPK self-efficacy or Technostress.
On the other hand, the conclusions section is quite long. It should be more concise
Yours faithfully
Author Response
Dear Madam or Sir,
Sincerest thanks to Guest Editor and reviewer for their valuable time and useful contribution.
The authors are grateful to all for careful reading of the paper and helpful suggestions and comments as a way to improve our work. The authors hope that a revised version of the manuscript will still be considered by International Journal of Environmental Research and Public Health.
The authors have modified the manuscript.
Below we provide our responses point by point and modify the manuscript in response to the accurate and intuitive the reviewer’ comments.
We added more explanation to the limitation section to clarify why the issues dedicated to Burnout are less than those dedicated to TPK self-efficacy or Technostress (See lines 757-766, corrected version of manuscript).
Accordingly, with suggestions we modified the future direction and conclusions (See lines 775-791).
Also, English language was revised.
We hope you find this revision appropriate for research topic. Once more, thank you for your exhaustive review.

Reviewer 3 Report
The abstract appears to contain an error as it references OC (rather than OS) as a significant but lower predictor. Congratulations on conducting such a timely study! The findings are not necessarily surprising or unique in that IM is positively associated with performance in many areas other than CI. Nonetheless, this confirms the importance of supporting IM with appropriate infrastructure, particularly given the necessary continuance of online instruction in pandemic times.
To what degree might exposure to others' online teaching models via social media be a factor in understanding IM and CI?
Under present circumstances, do instructors have the option of stopping online instruction, as suggested in the conclusion? Or were extrinsic motivators more prevalent than normal at the time of the study, continuing forward to the present time?
I do not see in the study the promise made in the abstract of valuable insights into how to "support teachers with coping during working remotely" - reading the claim "if coping strategies are formed, problems can be overcome in any crisis" begs the question. You might consider offering more specific suggestions in this or a future iteration of this work.
Author Response
Dear Madam or Sir,
Sincerest thanks to Guest Editor and reviewer for their valuable time and useful contribution.
The authors are grateful to all for careful reading of the paper and helpful suggestions and comments as a way to improve our work. The authors hope that a revised version of the manuscript will still be considered by International Journal of Environmental Research and Public Health.
The authors have modified the manuscript.
Below we provide our responses point by point and modify the manuscript in response to the accurate and intuitive the reviewer’ comments.
We modified the abstract. We have more clearly highlighted the relationship between IM and CI, so we added more studies to the introduction section to clarify this aspect. So, the introductory part has been fully modified.
Accordingly, with suggestions we modified the future direction and conclusions with ideas to support teachers with coping during working remotely.
Also, English language was revised.
We hope you find this revision appropriate for research topic. Once more, thank you for your exhaustive review.

Round 2
Reviewer 1 Report
Thank you for including the changes I suggested in the review. Authors need to make the introduction and literature crispy by avoiding unnecessary citations and checking for minor grammatical errors.